# Organo-Mineral Fertilizer Improves *Ocimum basilicum* Yield and Essential Oil

**DOI:** 10.3390/plants14070997

**Published:** 2025-03-22

**Authors:** Roberta Camargos de Oliveira, Mércia Freitas Alves, José Magno Queiroz Luz, Arie Fitzgerald Blank, Daniela Aparecida de Castro Nizio, Paulo César de Lima Nogueira, Sérgio Macedo Silva, Renata Castoldi

**Affiliations:** 1Agrarian Sciences Institute, Universidade Federal de Uberlândia, BR 050, km 78, Gloria Campus, Uberlandia 38410-337, Brazil; robertacamargoss@gmail.com (R.C.d.O.); jmagno@ufu.br (J.M.Q.L.); 2Agronomic Engineer Department, Universidade Federal de Sergipe, Avenue Marechal Rondon, São Cristóvão 49100-000, Brazil; merciafreitas.alvs@gmail.com (M.F.A.); arie.blank@gmail.com (A.F.B.); danielanizio82@gmail.com (D.A.d.C.N.); 3 Chemistry Department, Universidade Federal de Sergipe, Avenue Marechal Rondon, São Cristóvão 49100-000, Brazil; pclnogueira@uol.com.br; 4Agrarian Sciences Institute, Universidade Federal dos Vales do Jequitinhonha e Mucuri, Avenue Universitária, Unaí 38610-000, Brazil; sergio.macedo@ufvjm.edu.br; 5Agrarian Sciences Institute, Universidade Federal de Uberlândia, Campus Monte Carmelo, LMG-746, km 1, Monte Carmelo 38500-000, Brazil

**Keywords:** basil, sweet dani, cinnamon, aromatic plants, organo-mineral fertilizer, agricultural systems

## Abstract

The production of *Ocimum basilicum* (basil) crop depends upon the availability of all nutrients in the soil solution. There is a lack of information about its performance, at tropical conditions, using new fertilizer formulations, such as organo-mineral fertilizers, mainly under protected cultivation. These types of fertilizers combine benefits of the main fertilizers used in agriculture (organic and chemical). Therefore, organo-mineral fertilizers enhance soil health, provide a balanced nutrient supply, improve crop yields and quality. and promote environmental sustainability, making them a cost-effective and eco-friendly solution for sustainable crop production. This work aimed to evaluate the biomass and essential oil of basil varieties, with organo-mineral fertilization in different agricultural systems. Each experiment was conducted in a randomized block design, with three replications in a 2 × 4 factorial scheme, being two varieties of basil (“Sweet Dani” and “Cinnamon”) and four fertilizers: organo-mineral source, mineral source, organic source and the natural fertility of the soil. The evaluated characteristics were plant height, fresh biomass of plants, content, yield and the chemical composition of the essential oil. The organo-mineral sources of fertilizer provide better values for fresh biomass (average of 1175.90 and 1032.83 g per plant via greenhouse cultivation and field cultivation, respectively), essential oil yield (14.57 and 11.89 g per plant via greenhouse cultivation and field cultivation, respectively) and the dominant compounds for both cultivars of *O. basilicum*. Protected cultivation is the better environmental condition for obtaining the highest performance of *O. basilicum* cultivars about biomass and essential oil. The content of essential oil is not affected by the agricultural systems (greenhouse and field). The major compounds of essential oil under Brazilian crop conditions are Linalol and (E)-mehyl cinnamate in “Cinnamon” and neral and geranial (citral) in “Sweet Dani”.

## 1. Introduction

Basil is an important economic crop and is among the aromatic plants most investigated as a source of essential oils highly valued by the pharmaceutical and cosmetics industries [1].

Its characteristic aroma is due to its chemical composition, which has several organoleptic qualities. This species is much appreciated in traditional foods, sauces, and salads [2]. There are many varieties, genotypes and chemotypes of sweet basil, which vary in their leaf colors (green or purple), flower color (white, red, purple) and aroma [3].

The high economic value of basil oil is due to the presence of a complex mixture of volatile compounds, monoterpenes, sesquiterpenes and their oxygenated analogs present at low concentrations in plants [4]. The linalool, methyl chavicol, methyl eugenol, eugenol, and geraniol are major compounds of greater interest that characterize different basil varieties [5].

The chemical composition of an essential oil is affected mainly by the plant genotype, variety and chemotype, but exogenous factors also cause an important impact on the yield of these compounds [6], like water stress [7], salt stress [8], extraction processes [9] and drying methods [10].

Cultivation practices have an important influence on productivity and essential oil composition [11]. Previous studies have been particularly interested in how the soil conditions and fertilizers can provide better yields of essential oils from the species and its varieties [12]. Finding optimal fertilization sources for basil varieties could benefit growers and the environment. Understanding how to maximize basil’s growth (agricultural strategies) while minimizing resource use could lead to better crop management and support food security [13].

Many studies have presented satisfactory results regarding soluble fertilizers [14], organic materials [15,16] and the effects of combinations of them [17] on aromatic crop development and soil fertility management.

Soluble fertilizers provide readily available nutrients to the soil in a solution that can be easily assimilated by plants and generate a rapid response after application [18]. However, continuous use of inorganic fertilizer often leads to unsustainability, creating deficiencies of certain nutrients, acidification of the soil and other undesired effects. In response to this concern, there are worldwide concerted efforts to use green manuring, legumes, and organic manures to produce the same yield with less inorganic fertilizer [19].

Despite the low concentration of some elements, organic fertilizer provides unquestionable soil conditioning effects, and it is an indicator of natural fertility, producing improved soil conditions and greater presence of nutrients during longer periods of cultivation [20]. Like solutions to reuse waste and better conserve soil nutrients, new formulations of fertilizers at the junction of mineral and organic sources have improved potential crop production. Due to the presence of organic parts, organo-mineral fertilizers allow better retention of nutrients due to the high capacity of cation exchange [21].

The current trend is aimed at exploring the possibility of supplementing chemical with organic fertilizers that are eco-friendly and cost-effective [22]. However, knowing the needs of cultivars is essential for efficient nutrient management of agroecosystems. With balanced fertilization, crops can naturally provide more sustainability [23].

Particularly for some varieties of basil (“Sweet Dani” and “Cinnamon”), there is a lack of information about performance in tropical conditions using new fertilizer formulations, such as organo-mineral fertilizers, primarily under protected cultivation. Therefore, this work aimed to evaluate the biomass and essential oil yields of basil varieties with organo-mineral fertilization in different agricultural systems.

## 2. Results

### 2.1. Height, Fresh Biomass, Content and Yield of Essential Oil

The results, including plant height, fresh biomass, essential oil content and yield, are presented in Table 1 and Table 2. For the greenhouse, the data analysis shows that there was significant interaction between fertilizers and cultivars for fresh biomass, and the highest values occurred with the organo-mineral fertilizer for both cultivars. This same fertilizer also provided better results for essential oil yield. The taller plants were of the “Sweet Dani” cultivar using the organic fertilizer (117.66 cm).

In Table 2, which references the field experiment, the data analysis showed that plants of the “Sweet Dani” cultivar were significantly higher in the presence of the organo-mineral fertilizer. The organo-mineral fertilizer also provided significantly better results for fresh biomass, essential oil content and yield. For both studied areas, it is worth mentioning again that the values provided by mineral fertilization were lower than organic and organo-mineral fertilizers, in terms of plant height, biomass production and essential oil yield.

About the analysis of agricultural systems (Table 3 and Table 4), it is possible to observe that for the “Sweet Dani” cultivar, apart from plant height, the variables showed better results in the greenhouse than in the field. The same was not observed for the “Cinnamon” cultivar. For “Cinnamon”, the variables did not present significant differences between greenhouse and field, except for plant height. In this case, the better results were obtained in the greenhouse, but in general, the plants presented similar growth in both environments.

“Sweet Dani” showed a different performance than “Cinnamon”. Significant interactions were observed for height and fresh biomass. In this case, the greenhouse associated with organo-mineral fertilizer provided the better results for fresh biomass and tied with organic fertilizer for height. As for the essential oil yield, again the protected environment provided better conditions for a greater production of essential oil.

### 2.2. Essential Oil Composition

The results of measurements of the chemical composition of the essential oils for the two varieties of *Ocimum basilicum* L. are shown in Table 5 and Table 6. For “Sweet Dani”, 25 compounds were found, the most expressive being fenchyl, neral and geranial endo-acetate, composing more 75% of the essential oil. For “Cinnamon”, 21 compounds were found, being the major linalool, methyl chavicol and (E)- methyl cinnamate, constituting about 80% of the essential oil. Of all the main compounds, linalool was the dominant constituent of the essential oil.

For “Sweet Dani”, statistical analysis of the major compounds of the essential oil (Table 7 and Table 8) showed significantly better values for endo-acetate fenchyl using organic fertilization in the field. For neral, organic fertilization in the greenhouse led to higher percentages of compost. Finally, the geranial presented better values in the greenhouse when combined with organo-mineral fertilization.

For the “Cinnamon” cultivar, high percentages of the compounds were found using organo-mineral fertilizer and protected cultivation. However, particularly for (E)-methyl cinnamate, higher percentages were also obtained in the field when combined with organic fertilizer.

## 3. Discussion

### 3.1. Height, Fresh Biomass, Essential Oil Content and Yield

For both varieties of *O. basilicum* L., the characteristics of the organo-mineral fertilizer allowed us to obtain satisfactory results for biomass and essential oil yield in comparison to other fertilizers. An increase in the plant biomass and essential oil production of basil in two consecutive cultivation years (2013 and 2014) was observed by [24], in which organo-mineral fertilizer (carbonized biomass and chemical fertilizers) provided a nutrient supply for a longer period.

Other authors also found interesting results when they added organic fractions to nutrient management strategies [25,26]. As occurred in the work of [25], which showed higher essential oil yields when applying chemical fertilizer (41%), biofertilizer (33%) and combinations of biofertilizer and 50% chemical fertilizer (38%).

Our study is in discordance to [15], which obtained the highest fresh weight from the plants grown with conventional fertilizer at a rate of 250 kg N ha^−1^. For us, unsatisfactory results for both cultivars were obtained with exclusively mineral fertilization, not favoring fresh biomass and essential oil yield. In the early stages of crop development, there is still no fully formed root system for the absorption of nutrients already available in the soil solution. As a result, in the period of the greatest need for absorption, with greater metabolism by plants, the content of nutrients (especially N and K) in the soil is possibly lower. At that time, slow-release fertilizers have advantages for greater crop performance, specifically because such fertilizers break down over time, releasing nutrients gradually. This helps prevent the leaching of nutrients into the environment, which can otherwise harm nearby water systems. It also ensures that the nutrients are available to plants when they need them, rather than all at once [27].

Organic fertilizers have also led to satisfactory results for both cultivars. Organic sources, when compared to chemical fertilizers, have lower nutrient contents and are slow to release their nutrients, but are as effective as chemical fertilizers [28], especially for the activation and conservation of the soil microbiota responsible for the release of nutrients to plants.

The superiority of using the required N in biologic form and 75% in mineral form to improve vegetative growth may be due to the favorable effect of chemical nitrogen application on the activity of microorganisms responsible for biofertilizer decay in the soil, which increases the available N in soil and N-uptake, consequently encouraging the vegetative growth of the plant [26].

The protected cultivation showed that the sources of organo-mineral and organic fertilizers led to the best performance of “Sweet Dani” cultivar. Greenhouse cultivation favored the growth of plants because the plants were protected from adverse conditions of the climate and soil, evapotranspiration, solar radiation, the direct effects of rain, and leaching of nutrients. Our findings agree with [13], as the experiment was carried out in the rainy season. Plastic protection and controlled irrigation inside the greenhouse influenced the release of the nutrient supply from fertilizers, and in this case mineral fertilization took place more quickly due to its formulation. Furthermore, the high photosynthesis potential under organic fractions is probably due to the stimulated activity of beneficial soil microorganisms, enhancing the chlorophyll content of the plants [29].

Under field cultivation, organo-mineral and organic fertilizers had better production of fresh biomass and essential oil yield, probably because these fertilizers provided greater nutrient availability during the time of the experiments. The high soil humidity caused by the rainy period certainly led to greater leaching of the nutrients of more soluble sources, and organic formulations kept a larger quantity of nutrients around the root systems of the basil plants.

According to [30], combining organic and mineral fertilizers could result in increasing crop yields and quantities of antioxidant compounds (phenol and flavonoids) with fewer resources, including being able to reduce the rate of nutrient application without causing damage to production.

These results should be available to the extension section and pioneer farmers to consider as alternative nutrient sources in support of environmental health, especially in the production of high-quality crops such as medicinal plants.

Organic practices have been considered a good strategy considering the terms of economics, plant growth and environmental aspects, especially because by-products of cropping or animal production systems represent cheap, biodegradable and fully environmentally friendly materials for the production. Moreover, organic matter and plant residues can also decrease plants’ exposure to various pathogens and suppress some types of nematodes in the soil [31].

Organic fractions added to soils significantly improve the microbial community compared to the application of inorganic fertilizers alone. The presence of life in the soil strongly affects crop growth and soil quality, which has become an indicator of soil fertility and health. Because microorganisms are the dominant players in restoring soil functions and beneficial microorganisms can suppress soil-borne diseases, their presence enhances resistance to environmental stress [32].

The application of fractions of organic matter such humic acid was effective on the growth and yield parameters of field-grown plants, and the highest values were obtained from the highest dose of humic acid [33].

The benefits of this fertilizer segment bring new horizons to nutrient management, such as the application of organic fractions associated with nutrients and microorganisms, expanding the possibilities in management aimed at improving the development of basil varieties. These combinations can be very different, and the results reveal substantial increases, as reported by [34], who found root weight doubled or tripled compared to control plants when the plants received microorganisms associated with plant growth, chlorophyll and mineral uptake.

The essential oil and its chemical constituents vary drastically over the environment and season in the *Ocimum* plant [35,36,37]. It is always essential to determine the most suitable varieties for each region according to its climatic conditions. The variation between basil cultivars for herb growth and oil yield are usually related to genetic diversity due to hybridization, targeted cultivation, and breeding practices for desired morphochemotypes [38].

Regarding essential oil content, our findings are in accordance with [3], which studied the influence of nitrogen fertilizer on the performance of basil cultivars, observing essential oil contents between 0.8 and 2.07%.

Optimal and balanced mineral fertilization of aromatic plants, adjusted to their nutritional requirements and growing conditions, is an important cultivation factor determining the quantity and quality of the essential oil [39].

In our experiment, essential oil content presented similar results for both cultivars and agricultural systems, with small but not significant differences between fertilizers. Compost application on growing basil contributes to an increase in essential oil content [40]. According to [25], the highest essential oil content was obtained in chemical fertilizer and combinations of biofertilizer and 50% chemical fertilizer.

### 3.2. Essential Oil Composition

The constituents of basil essential oils across countries can be very different due to environmental and genetic factors, different chemotypes, harvest time, climate and the use of fertilizer. In northeast India, [41] reported camphor (42.1%), followed by limonene (7.6%) and β-selinene (5.6%), was the major component of the oil, while [42] found linalool (66.9–84.2%) and geranial (1.1–9.0%). In Turkey, methyl eugenol (78.02%) stood out against α-cubebene (6.17%) [43]. In China, [44] found 5-allylguaiacol, accounting for 50.2%, while [45] found methyl eugenol (12.93–25.93%) and eugenol (17.63–27.51%).

In [46], the authors identified seven chemotypes: linalool, methyl chavicol, citral/methyl chavicol, eugenol, methyl cinnamate/linalool, linalool/methyl eugenol, and methyl chavicol/linalool after characterizing the chemical diversity of Turkish basil (*O. basilicum* L.). Our study is in accordance with this study and [15], even though linalool was the dominant constituent for the “Cinnamon” cultivar.

The percentage of the dominant compounds linalool and geranial can also vary in relation to the season of harvest, despite the influence of the fertilizers and agricultural systems that are the objects of study in the present work. Knowledge regarding the occurrence of seasonal variability in the production of active ingredients is one of the main parameters to be considered in planning the harvest of medicinal and aromatic plants when the objective is the quality of the raw material and the presence of the active ingredients of interest [47].

In the present study, again, organic fertilizers provided better results for major compounds of essential oil. According to [48], organic fertilizer affects the essential oil composition in *O. basilicum* L. in a way similar to that of inorganic fertilization (ammonium nitrate). The authors of [3] related that the highest concentrations of the main constituents of essential oil (with the exception of methyl chavicol), namely 1,8-cineole and methyl cinnamate, were obtained under treatment without nitrogen fertilizer, as compared to treatments with nitrogen chemical fertilizers. For [40], compost application also increased the concentration of linalool and borneol in the oil, with a simultaneous decrease in the content of methyl chavicol and 1,8-cineole.

The higher rate of nitrogen fertilization causes an increase in volatile oil content and yield in some plants, as well as increasing the percentage of methyl chavicol and b-asarone and decreasing the linalool concentration in the oil [39]. The same author also reports that foliar feeding of nitrogen in the form of urea contributes to an increased concentration of linalool and epi-cadinol as well as a decreased content of 1,8-cineole, geraniol and eugenol in the oil of *O. basilicum*.

The different fertilizer sources improved micro-elements content due to enhancing the cation exchange capacity of the soil, the gradual release of nutrients, and the biological activities and physicochemical properties of the soil [49]. It is worth mentioning that the supply of micronutrients through organic fertilization possibly led to the higher production of compounds of interest in the essential oil, since nutrients such as zinc and manganese stimulate the route of production of secondary metabolites in secondary plant metabolism [13].

Increasing the efficiency of nutrient use in medicinal plants such as basil is a relevant factor because the metabolization of elements throughout development can affect the synthesis of secondary metabolite products, as reported by several researchers working with medicinal plants [50,51,52,53,54,55].

Due to the characteristics of the fertilizer (composition and combination with organic fractions), we observe that sources with organic components positively affect development. Therefore, the form of fertilization can affect the absorption of nutrients and their translocation in plants [56]. Working with *Echinacea purpurea* L., [57] observed maximum values in plant growth, total phenolics, total flavonoids and essential oil content in nutritional management with controlled-release fertilizer and greenhouse conditions. The results of this study suggested that the use of organo-mineral fertilizer has a beneficial effect on the biomass and essential oil composition of *O. basilicum* cultivars.

Foliar spraying of organic components during development has also been a subject of study by researchers, with encouraging results for the yield and production of medicinal plants [45]. Works like this and new formulations and forms of application are important to support and encourage new studies and new possibilities to supplement medicinal plants and make the valuable by-products of these species more accessible and of better quality.

## 4. Materials and Methods

### 4.1. Experimental Areas

The experiments were carried out at the Experimental Farm of Gloria, located in Uberlandia City, State of Minas Gerais, Brazil (18°57′ S e 48°12′ W), in two agricultural systems (field and greenhouse).

The experiments occurred during spring and summer, specifically April through June of 2017. According to the Koppen classification, the weather at the experimental area is Aw thermic, characterized by two seasons—a hot and humid summer and a dry and cold winter. The average total rain in the rainiest month is around 250 mm and the average annual total is between 1500 to 1600 mm.

For the protected environment, we used a high tunnel type greenhouse with the dimensions of 8 m wide, 50 m long and a 4 m side height, built with a metallic structure and covered with agricultural plastic film with a thickness of 150 μm.

### 4.2. Experimental Design

Each experiment was conducted in a randomized block design, with three replications in a 2 × 4 factorial scheme, using two varieties of basil (“Sweet Dani” and “Cinnamon”) and four treatments with fertilizers: organo-mineral source, mineral source, organic source and the natural fertility of the soil (control). The plots presented three lines with 10 plants each. The spacing was 60 cm between rows and 40 cm between plants.

The pelletized organo-mineral fertilizer was manufactured by Geociclo Biotechnology S/A, in Brazil. The compound was made of mineral nutrients, such as urea, monoammonium phosphate (MAP), and potassium chloride (KCl). Through an industrial process, it was transformed into an organo-mineral fertilizer. These fertilizers were produced using a biodegradable water-soluble organic polymer that gradually releases the macronutrients.

In the experiment, we used 250 kg ha^−1^ of organo-mineral fertilizer with the following characteristics: 4% total nitrogen; 14% P_2_O_5_; 8% total potassium (NPK 4-14-8); 8% organic carbon; 10% humidity; pH 6; density of 0.9 g cm^−3^; and a cation exchange capacity of 175.5 mmol kg^−1^. The mineral fertilizer was used by applying 250 kg ha^−1^ of the formulated NPK 4-14-8, with 4% total nitrogen (urea), 14% P_2_O_5_ (phosphate monoammonium), and 8% total potassium (potassium chloride).

The organic fertilizer was used by applying 4 t ha^−1^. The chemical composition of the cattle manure presented the following values: pH = 6.18; Nitrogen = 1.83%; Organic matter = 75.87%; Organic carbon = 28.79%; Relation C/N = 17/1; P_2_O_5_ = 0.56%; K_2_O = 1.95%; Calcium = 2.25%; Magnesium = 0.17%; Sulfur = 0.12%; Boron = 38 mg kg^−1^; Copper = 15 mg kg^−1^; Iron = 2798 mg kg^−1^; Manganese = 51 mg kg^−1^; Zinc = 48 mg kg^−1^ and Sodium = 447 mg kg^−1^.

### 4.3. Soil Analysis

According to the soil analysis, the experimental areas had the following characteristics: (1) Field: pH 6.2; P = 228 mg dm^−3^; K = 175 mg dm^−3^; Ca^2+^ = 4.5 cmol_c_ dm^−3^; Mg^2+^ = 1.5 cmol_c_ dm^−3^; Al^3+^ = 0 cmol_c_ dm^−3^; H+Al = 3.1 cmol_c_ dm^−3^; Sum of base = 6 cmol_c_ dm^−3^; cation exchange capacity = 11.23 cmol_c_ dm^−3^; base saturation = 68 %; organic matter = 8.9 dag kg^−1^; and organic carbon = 6.7 dag kg^−1^; (2) Greenhouse: pH 6.12; P = 234.67 mg dm^−3^; K = 273 mg dm^−3^; Ca^2+^ = 7.0 cmol_c_ dm^−3^; Mg^2+^ = 1.5 cmol_c_ dm^−3^; Al^3+^ = 0 cmol_c_ dm^−3^; H+Al = 2.0 cmol_c_ dm^−3^; SB = 8.5 cmol_c_ dm^−3^; T = 13.1 cmol_c_ dm^−3^; V = 77 %; OM = 7.0 dag kg^−1^; and OC = 5.8 dag kg^−1^. No corrective application was made in these areas.

### 4.4. Plant Material and Fertilizers Application

To obtain the seedlings, sowing of two varieties—“Sweet Dani” and “Cinnamon”—was done in trays using a conventional substrate. At 55 days after sowing, when the seedlings presented two or three pairs of true leaves, transplanting was performed to both areas. Soil preparation of the two areas was carried out 60 days before transplanting. The planting groove formation of the seedlings was made following the line spacing, and consequently the manual application of the fertilizers mentioned above (levels and sources). Before transplanting, the soil and fertilizer were mixed and then the seedling was placed in the prepared and fertilized soil.

The quantity of nutrients (N, P, and K) was based on the physical and chemical soil analyses and according to the literature that provides the nutrient levels used by basil [58].

A drip irrigation system was installed and set to a flow rate of 3.0 L h^−1^, spaced 0.75 × 0.75 m between rows of plants. Irrigation was applied daily during 45 min in the first days after transplanting for the establishment of the seedlings. During plant growth, irrigation was applied every two days for 60 min. The field had the same irrigation management, but during the rainy days, the irrigation was not carried out.

### 4.5. Harvests and Evaluations

Harvesting was performed at 105 days after planting in early summer, and only when more than 60% of the plants presented open flowers. Before this, the heights of the plants of the central line of each plot were obtained. Harvesting consisted of cutting the plants at about 20 cm above the ground. The fresh material was then weighed and stored for posterior analysis. The evaluated characteristics were plant height, fresh biomass of plants (flowers and leaves), content, yield and the chemical composition of the essential oil.

### 4.6. Extraction and Analysis of Essential Oil

Essential oils were obtained by the hydro distillation technique using a modified Clevenger apparatus. Each sample consisted of 150 g of fresh biomass (flowers and leaves in three repetitions of 50 g) distilled for 120 min. The essential oils were collected and stored in amber vials at −20 °C until the analysis of the chemical composition.

GC analyses of the essential oils were performed using gas chromatography coupled with mass spectrometry and flame ionization detection (GC-MS/FID; QP2010 Ultra, Shimadzu Corporation, Kyoto, Japan); the instrument was equipped with an autosampler AOC-20i (Shimadzu). Separations were accomplished using an Rtx^®^-5MS Restek-fused silica capillary column (5–diphenyl–95–dimethyl polysiloxane; 30 m × 0.25 mm i.d., 0.25 μm film thickness) at a constant helium (99.999%) flow rate of 1.2 mL min^−1^.

The essential oils were diluted in ethyl acetate and an injection volume of 0.5 μL (5 mg mL^−1^) was employed, with a split ratio of 1:10. The oven temperature was programmed to 50 °C (isothermal for 1.5 min), with an increase of 4 °C min^−1^ to 200 °C, then 10 °C min^−1^ to 250 °C, ending with a 5 min isothermal at 250 °C.

The MS and FID data were simultaneously acquired employing a Detector Splitting System; the split flow ratio was 4:1 (MS: FID). A 0.62 m × 0.15 mm i.d. restrictor tube (capillary column) was used to connect the splitter to the MS detector, and a 0.74 m × 0.2 mm i.d. restrictor tube was used to connect the splitter to the FID detector. The MS data (total ion chromatogram, TIC) were acquired in full scan mode (*m*/*z* of 40–350) at a scan rate of 0.3 scan/s using electron ionization (EI) with an electron energy of 70 eV.

The injector temperature was 250 °C and the ion-source temperature was 250 °C. The FID temperature was set to 250 °C and the gas supplies for the FID were hydrogen, air, and helium at flow rates of 30, 300 and 30 mL min^−1^, respectively. Quantification of each constituent was estimated by FID peak-area normalization (%). Compound concentrations were calculated from the GC peak areas, and they were arranged in order of GC elution.

Identification of individual components of the essential oil was performed by computerized matching of the acquired mass spectra with those stored in the NIST21, NIST107 and WILEY8 mass spectral libraries of the GC-MS data system. A mixture of hydrocarbons (C9H20–C19H40) was injected under these same conditions, and the identification of constituents was then performed by comparing the spectra obtained with those of the equipment data bank and according to the Retention Index calculated for each constituent as previously described [59]. Retention indices were obtained using the equation proposed by [60].

### 4.7. Statistical Analysis

The data were tested for residual normality assumptions (Shapiro–Wilk test) and for homogeneity between variances (Levene test). After this, treatment means were subjected to an analysis of variance (F test), followed by means tests (Tukey test). We also performed a joint analysis using the Tukey test considering the different agricultural systems.

## 5. Conclusions

The organo-mineral fertilizer provided better values for fresh biomass, essential oil yield and dominant compounds for both cultivars of *O. basilicum*.

During the rainy season, it is not recommended to use soluble fertilizer for the development and production of aromatic plants, because organic sources offer suitable and gradual quantities of nutrients for plant metabolism.

Protected cultivation is the better environmental condition for the highest performance of *O. basilicum* cultivars in the production of biomass and essential oil.

The contents of the essential oil are not affected by the choice of agricultural system (greenhouse and field).

The major compounds of essential oil produced in Brazilian crop conditions are Linalol and (E)-mehyl cinnamate in “Cinnamon” and neral and geranial (citral) in “Sweet Dani”.

## Figures and Tables

**Table 1 plants-14-00997-t001:** Means of plant height, fresh biomass, content and yield of essential oil of *Ocimum basilicum* L. cultivars at greenhouse cultivation.

Height (cm)					
Varieties	Organo-Mineral	Organic	Mineral	Control	Means
“Cinnamon”	86.33	106.00	93.66	96.33	95.58 b
“Sweet Dani”	123.33	129.33	130.00	130.00	128.16 a
Means	104.83 B	117.66 A	111.83 AB	113.16 AB	
F_value for cultivar_	0.000 *	F_value for fertilizer_	0.0219 *		
Fresh biomass (g per plant)					
“Cinnamon”	1160.49 aA	917.18 aB	618.40 bC	464.43 bD	790.12
“Sweet Dani”	1191.32 aA	979.00 aB	843.31 aC	763.66 aC	944.32
Means	1175.90	948.09	730.85	614.05	
F_value for cultivar_	0.000 *	F_value for fertilizer_	0.000 *		
Essential oil content (%)					
“Cinnamon”	1.21	1.08	1.15	0.89	1.098 a
“Sweet Dani”	1.26	1.15	1.05	0.96	1.097 a
Means	1.24 A	1.120 A	1.103 A	0.928 B	
F_value for cultivar_	0.9839 ns	F_value for fertilizer_	0.0009 *		
Essential oil yield (g per plant)					
“Cinnamon”	14.63	9.92	7.15	4.17	8.97 b
“Sweet Dani”	14.50	11.32	8.92	7.32	10.51 a
Means	14.57 A	10.62 B	8.03 C	5.75 D	
F_value for cultivar_	0.0014 *	F_value for fertilizer_	0.000 *		

Means followed by different uppercase letters in the line and lowercase letters in the column differ by the Tukey test (*p* < 0.05); * = significant; ns = not significant.

**Table 2 plants-14-00997-t002:** Means of plant height, fresh biomass, content and yield of essential oil of *Ocimum basilicum* L. cultivars in field cultivation.

Height (cm)					
Varieties	Organo-Mineral	Organic	Mineral	Control	Means
“Cinnamon”	87.66 bA	83.66 bA	79.00 bA	75.33 bA	83.41
“Sweet Dani”	112.66 aA	104.66 aAB	103.00 aB	83.33 aC	98.91
Means	100.16	94.16	91.00	79.33	
F_value for cultivar_	0.000 *	F_value for fertilizer_	0.000 *		
Fresh biomass (g per plant)					
“Cinnamon”	1037.50	853.33	724.19	610.89	806.48 a
“Sweet Dani”	1028.17	974.81	738.12	583.47	831.14 a
Means	1032.83 A	914.07 B	731.16 C	597.18 D	
F_value for cultivar_	0.2620 ^ns^	F_value for fertilizer_	0.000 *		
	Essential oil content (%)				
“Cinnamon”	1.13	1.03	0.96	1.03	1.048 a
“Sweet Dani”	1.16	1.09	1.09	0.99	1.078 a
Means	1.15 A	1.06 B	1.02 B	1.01 B	
F_value for cultivar_	0.1543 ^ns^	F_value for fertilizer_	0.0010 *		
Essential oil yield (g per plant)					
“Cinnamon”	12.07	8.78	6.98	6.30	8.53 a
“Sweet Dani”	11.70	10.62	8.07	5.79	9.04 a
Means	11.89 A	9.70 B	7.52 C	6.04 D	
F_value for cultivar_	0.0668 *	F_value for fertilizer_	0.000 *		

Means followed by different uppercase letters in the line and lowercase letters in the column differ by the Tukey test (*p* < 0.05); * = significant; ns = not significant.

**Table 3 plants-14-00997-t003:** Means of plant height, fresh biomass, content and yield of essential oil for “Cinnamon” cultivar, in function of agricultural systems and fertilizers.

Height (cm)					
Crop System	Organo-Mineral	Organic	Mineral	Control	Means
Field	87.66	83.66	79.00	83.33	83.41 b
Greenhouse	86.33	106.00	93.66	96.33	95.58 a
Means	87.00 A	94.83 A	86.33 A	89.33 A	
F_value for crop system_	0.002 *	F_value for fertilizer_	0.1029 ^ns^		
Fresh biomass (g per plant)					
Field	1037.50	853.33	724.19	610.89	806.48 a
Greenhouse	1160.49	917.18	618.40	464.43	790.12 a
Means	1098.99 A	885.25 B	671.33 C	537.66 D	
F_value for crop system_	0.3770 ^ns^	F_value for fertilizer_	0.000 *		
Essential oil content (%)					
Field	1.16	1.03	0.96	1.03	1.04 a
Greenhouse	1.26	1.08	1.15	0.89	1.09 a
Means	1.21 A	1.05 AB	1.05AB	0.96B	
F_value for crop system_	0.2541 ^ns^	F_value for fertilizer_	0.0071 *		
Essential oil yield (g per plant)					
Field	12.07	8.78	6.98	4.17	8.53 a
Greenhouse	14.63	9.92	7.15	6.30	8.97 a
Means	13.35 A	9.35 B	7.06 C	5.24 D	
F_value for crop system_	0.2030 ^ns^	F_value for fertilizer_	0.000 *		

Means followed by different uppercase letters in the line and lowercase letters in the column differ by the Tukey test (*p* < 0.05); * = significant; ^ns^ = not significant.

**Table 4 plants-14-00997-t004:** Means of plant height, fresh biomass, content and yield of essential oil for “Sweet Dani” cultivar, in function of agricultural systems and fertilizers.

Height (cm)					
Crop System	Organo-Mineral	Organic	Mineral	Control	Means
Field	112.66 bA	104.66 bA	103.00 bA	75.33 bB	98.21
Greenhouse	123.33 aA	129.33 aA	113.00 aB	92.25 aB	116.97
Means	118.00	117.00	108.50	83.66	
F_value for crop system_	0.000 *	F_value for fertilizer_	0.000 1*		
Fresh biomass (g per plant)					
Field	1028.17 bA	974.81 aA	738.12 bB	583.47 bC	831.14
Greenhouse	1191.32 aA	979.00 aB	843.31 aC	763.66 aC	944.32
Means	1109.74	976.90	790.71	673.57	
F_value for crop system_	0.0001 *	F_value for fertilizer_	0.000 *		
Essential oil content (%)					
Field	1.13	1.09	1.05	0.96	1.07 a
Greenhouse	1.21	1.15	1.09	0.99	1.09 a
Means	1.17 A	1.12 AB	1.07 AB	0.97 C	
F_value for crop system_	0.2978 ^ns^	F_value for fertilizer_	0.000 *		
Essential oil yield (g per plant)					
Field	11.70	10.62	8.07	5.79	9.04 b
Greenhouse	14.50	11.32	8.92	7.32	10.51 a
Means	13.10 A	10.97 B	8.49 C	6.55 D	
F_value for crop system_	0.0004 *	F_value for fertilizer_	0.000 *		

Means followed by different uppercase letters in the line and lowercase letters in the column differ by the Tukey test (*p* < 0.05); * = significant; ^ns^ = not significant.

**Table 5 plants-14-00997-t005:** General means of chemical compounds presents in “Sweet Dani” essential oil, at both agricultural systems.

			Greenhouse				Field			
Compounds	^1^ IRRC	IRRL	O	M	OM	C	O	M	OM	C
linalool	1080	1096	0.49	-	0.48	0.32	0.34	0.34	0.45	0.23
camphor	1130	1141	0.55	0.34	0.35	-	0.75	0.76	0.37	0.59
trans-verbenol	1144	1440	1.06	0.83	0.82	0.55	1.01	0.99	0.82	0.72
terpinen-4-ol	1161	1174	1.60	1.35	1.05	0.84	1.60	1.25	1.00	1.07
methyl chavicol	1180	1195	0.52	0.36	-	-	0.27	-	-	0.24
α-terpineol	1189	1186	0.39	0.30	0.25	0.16	0.60	0.39	0.28	0.37
nerol	1209	1227	1.91	1.69	4.57	3.77	1.56	2.58	4.49	1.82
endo-acetate fenchyl	1216	1218	7.63	5.39	6.49	4.12	9.93	9.17	7.08	8.08
neral	1224	1235	28.14	25.13	23.09	20.58	24.87	26.13	24.45	26.62
geraniol	1234	1249	1.50	1.71	-	-	1.67	1.51	-	1.69
trans-sabinene hydrate acetate	1248	1253	2.76	2.46	-	-	3.13	3.08	-	3.31
geranial	1253	1264	32.22	33.12	33.54	39.15	29.81	31.33	32.38	33.11
δ-elemene	1338	1335	0.48	0.43	0.34	0.45	0.44	0.44	0.33	0.55
eugenol	1357	1356	0.87	1.06	0.95	0.96	0.83	0.78	1.01	0.75
(E)- methyl cinnamate	1372	1376	0.99	1.43	1.25	1.25	0.99	0.87	1.32	0.85
trans-β-copaene	1406	1419	3.96	5.22	5.70	5.73	4.29	4.15	6.03	4.07
α-cis-bergamotene	1416	1411	1.27	1.57	1.88	2.01	1.33	1.31	1.57	1.31
α-guaiene	1421	1437	0.32	0.53	0.39	0.49	0.31	0.31	0.43	0.33
α-humulene	1440	1452	0.68	1.02	1.01	1.13	0.76	0.69	1.01	0.68
germacrene D	1466	1484	3.12	4.77	4.85	4.84	3.34	3.34	3.97	3.32
β-selinene	1474	1489	2.34	2.82	3.52	3.54	2.88	2.49	3.33	2.48
bicyclogermacrene	1481	1500	1.85	2.31	3.37	2.92	2.14	2.02	3.49	1.98
γ-cadinene	1502	1513	0.24	0.37	0.43	0.32	0.26	0.25	0.46	0.32
β-sesquiphellandrene	1520	1521	2.98	3.89	4.18	4.55	3.42	3.57	4.29	3.48
longipinanol	1578	1569	1.18	1.04	1.15	1.48	1.63	1.08	1.23	1.29

^1^ IRRC: Relative Retention Index—calculated; IRRL: Relative Retention Index—literature; O: Organic, M: Mineral, OM: Organo-mineral; C: Control.

**Table 6 plants-14-00997-t006:** General means of chemical compounds presents in “Cinnamon” essential oil at both agricultural systems.

			Greenhouse				Field			
Compounds	^1^ IRRC	IRRL	O	M	OM	C	O	M	OM	C
1,8-cineol	1015	1026	4.29	3.77	4.19	3.38	2.24	2.22	3.57	1.86
(Z)-z-β-ocimene	1028	1032	0.88	0.63	0.48	0.60	-	-	0.43	0.26
γ-terpinene	1050	1054	0.45	0.50	-	0.53	0.36	0.35	0.36	0.35
linalool	1087	1095	34.57	34.42	48.13	34.42	37.34	37.35	42.11	35.22
camphor	1130	1141	0.65	0.47	-	0.41	0.37	0.37	0.50	0.37
terpinen-4-ol	1161	1174	1.25	1.25	1.86	1.48	1.06	1.03	1.01	1.08
α-terpineol	1174	1186	0.37	0.37	-	0.37	0.25	0.24	-	-
methyl chavicol	1185	1195	9.47	9.17	19.11	18.77	10.97	9.17	15.49	5.42
isobornyl acetate	1267	1283	0.26	0.29	0.38	0.22	0.35	0.34	0.21	0.31
(Z) methyl cinnamate	1288	1299	3.63	2.98	1.46	4.87	2.68	2.65	2.39	3.20
(E)- methyl cinamato	1372	1376	32.06	27.31	35.48	32.90	37.63	33.69	37.66	9.03
α-cis-bergamotene	1416	1411	1.08	1.28	1.95	0.63	0.58	0.57	0.90	1.05
α-guaiene	1421	1437	0.43	0.51	0.49	0.52	0.42	0.46	0.37	0.50
α-humulene	1440	1452	0.23	0.32	0.34	0.33	0.23	0.23	0.23	0.32
germacrene D	1466	1484	1.66	1.82	2.18	1.93	1.26	1.31	1.93	2.11
bicyclogermacrene	1481	1500	0.35	0.42	0.65	0.55	0.33	0.33	0.39	0.38
α-bulnesene	1488	1509	0.55	0.75	0.81	0.81	0.65	0.65	0.55	0.77
γ-cadinene	1497	1513	1.19	1.39	1.62	1.61	2.05	2.02	1.28	1.69
(E)-nerolidol	1538	1531	0.23	0.43	0.48	0.53	0.62	0.63	0.50	0.66
1,10-di-epi-cubenol	1598	16118	0.46	0.48	0.57	0.50	0.62	0.64	0.54	0.78
α-epi-cadinol	1625	1638	2.86	2.83	3.76	3.27	3.33	3.35	2.89	3.78

^1^ IRRC: Relative Retention Index—calculated; IRRL: Relative Retention Index—literature; O: Organic, M: Mineral, OOM: Organo-mineral; C: Control.

**Table 7 plants-14-00997-t007:** Means (%) of major compounds present in essential oil of the “Sweet Dani” (*Ocimum basilicum* L.) under interaction between agricultural systems and fertilizers.

		Endo-Acetate Fenchyl			
Crop System	Organo-Mineral	Organic	Mineral	Control	Means
Field	7.07 aD	9.93 aA	9.17 aB	8.08 aC	8.56
Greenhouse	6.48 bB	7.63 bA	5.39 bC	5.59 bC	6.27
Means	6.78	8.78	7.28	6.83	
Fvalue for interaction	0.000 *				
		neral			
Field	24.45 aB	24.86 bB	26.13 aA	20.14 aC	23.89
Greenhouse	23.06 bC	28.14 aA	25.13 aB	20.58 aD	24.23
Means	23.76	26.50	25.63	20.36	
F_value for interaction_	0.000 *				
		geranial			
Field	32.38	29.81	31.33	28.86	30.59 B
Greenhouse	33.54	32.22	33.11	31.39	32.56 A
Means	32.96 A	31.01 B	32.23 A	30.12 B	
F_value for interaction_	0.2265 ^ns^				

Means followed by different uppercase letters in the line and lowercase letters in the column differ by the Tukey test (*p* < 0.05); * = significant; ^ns^ = not significant.

**Table 8 plants-14-00997-t008:** Means (%) of major compounds present in essential oil of the “Cinnamon” variety (*Ocimum basilicum* L.) under interaction between agricultural systems and fertilizers.

		Linalool			
Ag. System	Organo-Mineral	Organic	Mineral	Control	Means
Field	42.11 bA	37.34 aB	37.35 bB	35.22 bC	38.00
Greenhouse	48.13 aA	34.57 bC	40.70 aB	34.42 aD	39.45
Means	39.02	35.95	39.02	34.82	
F_value for interaction_	0.000 *				
		(E)-methyl cinnamate			
Field	35.48 bB	37.63 aA	23.69 bC	19.02 aD	29.50
Greenhouse	37.66 aA	32.06 bB	27.30 aC	12.90 bD	26.93
Means	36.57	34.84	25.49	15.96	
F_value for interaction_	0.000 *				
		methyl chavicol			
Field	15.49 bA	10.97 aB	9.17 aC	5.42 bD	10.26
Greenhouse	19.11 aA	9.47 bB	9.17 aB	11.77 aA	14.13
Means	17.30	10.22	9.17	8.09	
F_value for interaction_	0.000 *				

Means followed by different uppercase letters in the line and lowercase letters in the column differ by the Tukey test (*p* < 0.05); * = significant.

## Data Availability

Data will be made available on request.

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
