# Peer review of "Organo-Mineral Fertilizer Improves Ocimum basilicum Yield and Essential Oil"

_plants, 2025, doi:10.3390/plants14070997_

Round 1

Reviewer 1 Report (New Reviewer)

Comments and Suggestions for Authors

The msnudcript "Organo-mineral fertilizer improves Ocimum basilicum yield and essential oil" examines the impact of different fertilizers on the biomass and essential oil production of two basil cultivars (“Sweet Dani” and “Cinnamon”) under tropical conditions and protected cultivation. Conducted using a randomized block design, it tested four fertilizer types: organo-mineral, mineral, organic, and natural soil fertility. Results showed that organic fertilizers led to higher fresh biomass and essential oil yield for both varieties. Protected cultivation improved overall plant performance, though essential oil content remained unchanged across agricultural systems. The dominant essential oil compounds were linalool and (E)-methyl cinnamate in cv. “Cinnamon” and neral and geranial (citral) in cv. “Sweet Dani.”

Comment: Instead of variety, use cultivar. Put the cultivar name in quotation marks and correct it throughout the text.

Comment: Lines 43-44: instead cultivar use variety, genotype and chemotype!

Comment: Line 47: You start the sentence with: [5] has reported that linalool, methyl chavicol, methyl -... Paraphrase a sentence, you can't start a sentence like that.

Comment: Line 56: You say ... plant genotype,... yes, great, add the variety and chemotype as well.

Comment: Tables 1&2: instead variety use cultivar!

Comment: Lines 287-301: ??? Delete!!!

Comment: provide more information about fertilizers used in this study!

Author Response

Dear Revieew,

Thank you for providing helpful suggestions to improve the manuscript entitled "Organo-mineral fertilizer improves Ocimum basilicum yield and essential oil".

In the attached files, we addressed the comments raised by the reviewers and did our best to implement all the suggestions. All comments and suggestions, along with our responses, are listed below and are also denoted in “red color” in the edited manuscript. We hope that we were able to answer all the questions and that the manuscript is now suitable for publication in Plants.

Sincerely, 

Renata Castoldi

 1) Comment: Instead of variety, use cultivar. Put the cultivar name in quotation marks and correct it throughout the text.

Response:  We have changed variety to cultivar throughout the text and put the cultivar name in quotation marks and correct it throughout the text.

2) Comment: Lines 43-44: instead cultivar use variety, genotype and chemotype!

Response: We use cultivar use variety, genotype and chemotype.

3) Comment: Line 47: You start the sentence with: [5] has reported that linalool, methyl chavicol, methyl -... Paraphrase a sentence, you can't start a sentence like that.

Response: We modified all the prayers that were in this way.

4) Comment: Line 56: You say ... plant genotype,... yes, great, add the variety and chemotype as well.

Response: We add variety and chemotype as well.

5) Comment: Tables 1&2: instead variety use cultivar!

Response: We agree and fix.

6) Comment: Lines 287-301: ??? Delete!!!

Response: We delete.

7) Comment: provide more information about fertilizers used in this study!

Response: Fertilizer descriptions are in lines 371 to 381.

Reviewer 2 Report (New Reviewer)

Comments and Suggestions for Authors

The paper described a study of the organo-mineral fertilizer upon the yield and essential oil composition of basil. The study is well structured and gives relevant insights into the effects of different fertilization strategies under field and greenhouse conditions. Nevertheless, need to be improved for clarity, scientific soundness, and impact.

The statistical analyses should be described in more detail. It is unclear whether multiple comparisons were corrected for statistical significance.

Some sections, especially the Introduction, lack logical flow. There is excessive background information that could be reduce for conciseness.

Figures and tables should be explicitly referenced in the text as evidence for findings.

Change volatile substances to …. volatile organic compounds (VOCs)

Also majorities compound….. major compounds

Discussion on fertilization and soil nutrients has been repeated many times , please reduce it.

Clarify how the study contributes to the field and why it is important  

212-214 If organic fertilizers have lower nutrient content, in what way are they "as effective"? Are they equally effective for plant growth, or just for soil health?

Line 268-270 as compared to its nitrogen chemical fertilizers....... is unclear. Do you mean as compared to treatments with nitrogen chemical fertilizers?

Line 201 algo… typo "also"

Line 283 benefic….. typo beneficial

How were the selection criteria for the cultivars Sweet Dani and Cinnamon basil and how did the oil yield percentages from those selections compare with those of other commercially available varieties? 

What were the light intensity and temperature conditions of the greenhouse?

 Why was Tukey's test preferred over other alternatives, such as Duncan's multiple range test?

What is the rationale for choosing a 120-minute hydro-distillation time? Rationale of this timeframe did it in fact maximize the yield of oil?

What are the effects upon long-term soil health of continuously using organo-mineral fertilizers in the commercial cultivation of basil?

Discussion lacks depth in comparing results with previous studies.

Author Response

Dear Revieew,

Thank you for providing helpful suggestions to improve the manuscript entitled "Organo-mineral fertilizer improves Ocimum basilicum yield and essential oil".

In the attached files, we addressed the comments raised by the reviewers and did our best to implement all the suggestions. All comments and suggestions, along with our responses, are listed below and are also denoted in “red color” in the edited manuscript. We hope that we were able to answer all the questions and that the manuscript is now suitable for publication in Plants.

Sincerely, 

Renata Castoldi

 1) Some sections, especially the Introduction, lack logical flow. There is excessive background information that could be reduce for conciseness.

Response:  We appreciate your feedback and have made sure to remove the excessive background information to improve clarity.

2) Figures and tables should be explicitly referenced in the text as evidence for findings. ???

Change volatile substances to …. volatile organic compounds (VOCs)

Response: It was corrected in the text, thanks for the comment

3) Also majorities compound….. major compounds

Response: It was corrected in the text, thanks for the comment

4) Discussion on fertilization and soil nutrients has been repeated many times , please reduce it.

Response: Thank you for your thoughtful comment. We appreciate your feedback and have made sure to remove the repetitions regarding fertilization and soil nutrients to improve clarity.

5) Clarify how the study contributes to the field and why it is important  

Response: By finding the optimal fertilization sources for basil, this research brings benefit for growers and the environment. Basil is widely used in culinary and medicinal applications, understanding how to maximize its growth (agricultural strategies) while minimizing resource use can lead to better crop management and support food security (Gavrić et al., 2021).

6) 212-214 If organic fertilizers have lower nutrient content, in what way are they "as effective"? Are they equally effective for plant growth, or just for soil health?

Response: It is because the nutrientes have slow-release (Slow-release fertilizers break down over time, releasing nutrients gradually. This helps prevent the leaching of nutrients into the environment, which can otherwise harm nearby water systems. It also ensures that the nutrients are available to plants when they need them, rather than all at once).

7) Line 268-270 as compared to its nitrogen chemical fertilizers....... is unclear. Do you mean as compared to treatments with nitrogen chemical fertilizers?

Response: We correct in text to clarify, thanks.

8) Line 201 algo… typo "also"

Response: We correct in text.

9) Line 283 benefic….. typo beneficial

Response: We correct in text.

10) How were the selection criteria for the cultivars Sweet Dani and Cinnamon basil and how did the oil yield percentages from those selections compare with those of other commercially available varieties? 

Response: The varieties were chosen due to their adaptability and productivity (performance) in the region where they were developed (unpublished data).

11) What were the light intensity and temperature conditions of the greenhouse?

Response: Unfortunately, we do not have these informations because there was no equipment to measure these datas in our greenhouse. We appreciate your kind feedback and will provide ways for measurements to be taken in future researches.

 12) Why was Tukey's test preferred over other alternatives, such as Duncan's multiple range test?

Response: Duncan’s test might be more useful in some cases. However,  often have a higher risk of some errors. For this reason we chosen Tukey’s test: we considered the test more reliability, conservative to control of error rates, and bring more comprehensive approach to comparing group means, considering the integrity of the results a priority.

13) What is the rationale for choosing a 120-minute hydro-distillation time? Rationale of this timeframe did it in fact maximize the yield of oil?

Response: After 120 minutes there is no increase in the amount of EO extracted and on the other hand there is a risk of burning the EO. This information can be clarified in: EHLERT, P.A.D.; BLANK, A.F.; ARRIGONI-BLANK, M.F.; PAULA, J.W.A.; CAMPOS, D.A.; ALVIANO, C.S.Tempo de hidrodestilação na extração de óleo essencial de sete espécies de plantas medicinais. Rev. Bras. Pl. Med., Botucatu, v.8, n.2, p.79-80, 2006.

14) What are the effects upon long-term soil health of continuously using organo-mineral fertilizers in the commercial cultivation of basil?

Response: We appreciate the kind comment and we have reworked it to improve the comparison between the results found in this work and those available in the literature.

15) Discussion lacks depth in comparing results with previous studies.

Response: We improve the discussion.

Reviewer 3 Report (New Reviewer)

Comments and Suggestions for Authors

Dear Authors

My comments on your article are stated below.

Abstract Part:

- In the abstract, it would be more appropriate to write the Latin name of the plant first and then write the basil in parentheses.

- The importance of organo-mineral fertilizers can be emphasized with a sentence in the abstract.

 - Important results should be given numerically, even in parentheses.

- It should be stated which variety is prominent in the abstract.

Materials and Methods Part:

- 4. The description under the material method section should be deleted

- Please indicate between which months the greenhouse and field trials were conducted.

- Climatic data such as average temperature, precipitation and humidity of the trial area should be given. Especially the climate values ​​of the months in which the trial is conducted should be stated.

- Is there a temperature and humidity measuring device inside the greenhouse? It would be good if the temperature and humidity values ​​inside the greenhouse were given.

-The distance between rows and on rows is too much. It would be good if a reference is given.

-How were fertilizer doses determined? Please provide references

-Were fertilizers given while the seedlings were being transplanted?

-Was there only one harvest in basil varieties? At least two harvests can be done in basil.

Results Part:

- The results obtained from the examined traits should be given in a certain order. For example, the plant height values ​​of the Sweet Dani variety varied between 123.33-130 cm, and the highest plant height was obtained in the mineral and control applications in greenhouse conditions. In this way, the values ​​of the same feature examined under field conditions can be given. I think it will be more understandable this way.

Discussion Part:

-In the discussion section, studies on basil and different organic-chemical fertilizers should be given. There are very valuable studies on basil and different fertilizer applications. There are very superficial and general statements in the discussion.

-There are too many sentences starting with the reference number. This situation negatively affects fluency. The reference number can be used with connecting sentences or at the end of the sentence.

-There are word errors in the article. Please check it. For example [26] algo found highest fresh and dry weights and essential oil content using NPK (75%) + biological fertilizers

Conclusion part :

- The conclusion section is very insufficient. It should be stated which variety and which fertilizer application are prominent in the examined traits. Also, important results related to greenhouse and field conditions should be emphasized.

Best regards,

Author Response

Dear Revieew,

Thank you for providing helpful suggestions to improve the manuscript entitled "Organo-mineral fertilizer improves Ocimum basilicum yield and essential oil".

In the attached files, we addressed the comments raised by the reviewers and did our best to implement all the suggestions. All comments and suggestions, along with our responses, are listed below and are also denoted in “red color” in the edited manuscript. We hope that we were able to answer all the questions and that the manuscript is now suitable for publication in Plants.

Sincerely, 

Renata Castoldi

 Abstract Part

1) In the abstract, it would be more appropriate to write the Latin name of the plant first and then write the basil in parentheses.

Response:  We agree and fix.

2) The importance of organo-mineral fertilizers can be emphasized with a sentence in the abstract.

Response: We agree and fix.

3) Important results should be given numerically, even in parentheses.

Response: We agree and fix.

4) It should be stated which variety is prominent in the abstract. 

Response: The Abstract states that there was no difference between the cultivars.

Materials and Methods Part:

1) The description under the material method section should be deleted 

Response: We agree and fix.

2) Please indicate between which months the greenhouse and field trials were conducted.

Response: We indicated.

3) Climatic data such as average temperature, precipitation and humidity of the trial area should be given.

Especially the climate values ​​of the months in which the trial is conducted should be stated.

Is there a temperature and humidity measuring device inside the greenhouse? It would be good if the temperature and humidity values ​​inside the greenhouse were given.

Response:  Unfortunately, we do not have these informations because there was no equipment to measure these datas in our greenhouse. We appreciate your kind feedback and will provide ways for measurements to be taken in future researches.

4) The distance between rows and on rows is too much. It would be good if a reference is given.

Response: More space (60 cm between rows and 40 cm within rows) is a part of an experimental design to compare plant growth and yield. Although wider spacing may reduce the number of plants per unit area, the overall health and productivity of each plant could result in a higher quality yield, making it a cost-effective choice in the long run.  Wider spacing allows for better airflow between the plants, which helps to reduce humidity around the plants. This can lower the risk of fungal diseases and other plant pathogens, which are more likely to thrive in tightly packed environments with poor ventilation. The basil plants receive better light exposure, especially in the lower parts of the plant. This can promote stronger growth, better leaf production, and higher quality foliage. Other benetifts can include: less competition for nutrients and water, easier harvesting and maintenance the area.

The space used was recommended by Pereira, Rita de Cassia Alves in Manjericão: cultivo e utilização / Rita de Cassia Alves Pereira, Ana Luzia Martins Moreira. – Fortaleza: Embrapa Agroindústria Tropical, 2011. 31 p.; I. 21 cm. – (Documentos / Embrapa Agroindústria Tropical, ISSN 2179-8184, 136).

5) How were fertilizer doses determined? Please provide references

Response: We provide.

6) Were fertilizers given while the seedlings were being transplanted?

Response: The planting groove formation of the seedlings was made following the line spacing, and consequently, the manual application of the fertilizers mentioned above (levels and sources). Before transplanting, the soil and fertilizer were mixed and then the seedling was placed in the prepared and fertilized soil.

7) Was there only one harvest in basil varieties? At least two harvests can be done in basil.

Response: We make just one harvest in this work.

Results Part:

1) The results obtained from the examined traits should be given in a certain order. For example, the plant height values ​​of the Sweet Dani variety varied between 123.33-130 cm, and the highest plant height was obtained in the mineral and control applications in greenhouse conditions. In this way, the values ​​of the same feature examined under field conditions can be given. I think it will be more understandable this way.

Response: We agree and fix.

Discussion Part:

1) In the discussion section, studies on basil and different organic-chemical fertilizers should be given. There are very valuable studies on basil and different fertilizer applications. There are very superficial and general statements in the discussion.

Response: We added about 30 references and discussed them

2) There are too many sentences starting with the reference number. This situation negatively affects fluency. The reference number can be used with connecting sentences or at the end of the sentence.

-There are word errors in the article. Please check it. For example [26] algo found highest fresh and dry weights and essential oil content using NPK (75%) + biological fertilizers

Response: We agree and fix.

Conclusion part :

1) The conclusion section is very insufficient. It should be stated which variety and which fertilizer application are prominent in the examined traits. Also, important results related to greenhouse and field conditions should be emphasized.

Response: We report in conclusion that there was no difference between the cultivars, we placed the cultivation environment that provides better production characteristics and essential oils.

Round 2

Reviewer 1 Report (New Reviewer)

Comments and Suggestions for Authors

The authors have addressed all comments correctly and incorporated them into the text. I propose that the manuscript Organo-mineral Fertilizer Improves Ocimum basilicum Yield and Essential Oil be accepted for publication in Plants in its current form.

Author Response

Dear reviewer

Thank you for your valuable suggestions.

We are sure that they will be important for improving the article.

Reviewer 2 Report (New Reviewer)

Comments and Suggestions for Authors

Authors significantly improved MS and I recommend it to accept now. 

Author Response

Dear reviewer

Thank you for your valuable suggestions.

We are sure that they will be important for improving the article.

Kind regards

Renata Castoldi

Reviewer 3 Report (New Reviewer)

Comments and Suggestions for Authors

Dear Author

In general, the requested corrections have been made. But I would like to state that I do not fully agree with your opinion regarding planting distance.

In field experiments, planting density is usually determined by considering the morphological development of the plant and the characteristics such as yield and disease that you have specified. However, leaving the planting density distance wider than necessary indicates that the field area is not used correctly and therefore the yield obtained from the unit area will be low. I think it is important to adjust the planting distance correctly since the planting-planting soils are limited.

Best regards,

Author Response

Dear reviewer

I believe your opinion is relevant, however, it is not possible to change the spacing in the text, since the experiment has already been implemented.

But your suggestions were valuable and I believe we will use them in future experiments.

This manuscript is a resubmission of an earlier submission. The following is a list of the peer review reports and author responses from that submission.

Round 1

Reviewer 1 Report

Comments and Suggestions for Authors

Many details need to be refined.
